

# Reproductive strategies in loggerhead sea turtle *Caretta caretta*: polyandry and polygyny in a Southwest Atlantic rookery

Laís Amorim[1], Lara Chieza[1], Jake A. Lasala[2], Sarah de Souza Alves Teodoro[1], Wesley D. Colombo[1], Ana Carolina Barcelos[1], Paula Rodrigues Lopes Guimarães[1], João Luiz Guedes da Fonseca[1], Ana Claudia Jorge Marcondes[3], Alexsandro Santos[3] and Sarah Vargas[1]

[1] Departamento de Ciências Biológicas, Universidade Federal do Espírito Santo, Laboratório de Genética e Evolução Molecular, Vitória, Espírito Santo, Brazil
[2] Sea Turtle Conservation Research Program, Sarasota, Florida, United States
[3] Fundação Projeto Tamar, Vitória, Espírito Santo, Brazil

## ABSTRACT

Sea turtles are highly migratory and predominantly inhabit oceanic environments, which poses significant challenges to the study of their life cycles. Research has traditionally focused on nesting females, utilizing nest counts and mark-recapture methods, while male behavior remains understudied. To address this gap, previous studies have analyzed the genotypes of females and hatchlings to indirectly infer male genotypes and evaluate the extent of multiple paternity within populations. Our research aimed to investigate the presence of multiple paternity in loggerhead turtle (*Caretta caretta*) nests for the first time in Brazil. We analyzed 534 hatchlings from 43 nests associated with 42 females during the 2017/18 to 2019/20 nesting seasons, using four highly polymorphic, species-specific microsatellite markers (nDNA). Parentage tests were conducted to reconstruct paternal genotypes and determine the rates of multiple paternity within clutches. Our results revealed that 72.09% of clutches were sired by multiple males, with contributions ranging from one to six males per clutch. Additionally, seven out of 88 males (7.95%) were found to have sired clutches from multiple females, with some males contributing to more than one clutch within and across breeding seasons. The breeding sex ratio (BSR) was calculated to be 2.09 males per female. While multiple paternity is a common phenomenon among sea turtles, this study is the first to document polyandry in loggerheads in Brazil and the first to provide evidence of polygyny in this species globally. This research establishes a crucial database for future studies in Brazil, with a focus on the BSR of the Southwest Atlantic subpopulation, offering essential insights for developing effective management strategies for this vulnerable population.

Corresponding author
Sarah Vargas, sarah.vargas@ufes.br

## INTRODUCTION

Sea turtles play a vital role in global biodiversity maintenance. While some populations can grow significantly, few remain unaffected by human activities (*Marcovaldi, dos Santos & Sales, 2011*). Observing sea turtles is difficult since these animals are both littoral and pelagic (*Moore & Ball, 2002*). Consequently, many aspects of their biology and behavior, particularly those of males, remain unclear. Males stay in the ocean, hindering observations, whereas females return to their natal regions to nest, providing ample research opportunities (*Bowen et al., 1993*). Both sexes migrate to mating areas before nesting, with males contributing to gene flow and female natal philopatry, shaping regional genetic structure (*Bowen et al., 1993*; *FitzSimmons et al., 1997*; *Reid et al., 2019*). However, most studies focus on females using nest counts and mark-recapture data (*Casale et al., 2023*; *Lasala et al., 2023*), leaving male behavior underexplored.

The loggerhead sea turtle (*Caretta caretta*) is listed as "Vulnerable" on both the Brazilian Red List (*Brasil, 2022*) and the IUCN Red List (*Casale & Tucker, 2017*). Brazil hosts one of the largest remaining loggerhead nesting populations in the world, second only to the super-aggregations found in Oman, and eastern Florida-USA (*Marcovaldi & Chaloupka, 2007*). In Brazil, loggerhead nesting sites are primarily located along the northeast (Sergipe and Bahia) and southeast coast (Espírito Santo and Rio de Janeiro) (*Marcovaldi, dos Santos & Sales, 2011*; *Marcovaldi et al., 2016*). These populations are typically defined by the geographic locations of these key nesting sites along the coast.

In Espírito Santo, long-term nesting female loggerheads exhibit strong nest site fidelity, meaning they repeatedly return to the same nesting sites (*Barreto et al., 2019*). This behavior is a well-documented characteristic of the species (*Miller, 1997*) and provides a reliable opportunity for researchers to study and monitor these turtles in a consistent environment. Espírito Santo, therefore, serves as an ideal location for conducting long-term studies on loggerhead nesting behavior, population dynamics, and conservation efforts. The high fidelity to nesting sites not only facilitates the sampling of individuals but also enhances the effectiveness of conservation programs aimed at protecting these critical habitats.

Loggerheads reach sexual maturity between 25 and 35 years old (*Chaloupka & Limpus, 1997*; *Chaloupka & Musick, 1997*). Hormone analysis shows males can breed successively within a breeding season and potentially in consecutive years (*Wibbels et al., 1990*), and satellite tracking data confirms they stay close to breeding grounds (*Hays, Mazaris & Schofield, 2014*). Males likely mate more frequently than females due to lower reproductive investment. Post-mating, males return to foraging grounds, while females travel to nesting beaches (*Bowen et al., 2004*), laying up to seven clutches per season and return every 2 to 3 years (*Dodd, 1988*). In Brazil, however, the northern Brazilian loggerheads lay, on average, 3.1 clutches per season and return every 2.4 years (*Marcovaldi et al., 2010*).

The sex of sea turtles is determined by the temperature of their nests, otherwise known as temperature-dependent sex determination (TSD) (*Montero et al., 2018*). Simply put, warmer nest temperatures tend to produce more females, while cooler temperatures result in more male hatchlings (*Marcovaldi, Godfrey & Mrosovsky, 1997*). In addition, other
factors include moisture level (*Lolavar & Wyneken, 2015*; *Montero et al., 2018*). This temperature-dependent mechanism is very important to understand the development and adaptation of sea turtle populations, as it directly influences the sex ratio in a population.

The pivotal temperature for a 1:1 sex ratio in loggerheads is around 29.3 °C (*Mrosovsky et al., 2002*), with a similar value of 29.2 °C recorded in Bahia, Brazil (*Marcovaldi, Godfrey & Mrosovsky, 1997*). In the southern regions, a near-balanced sex ratio of 53% female was found in Espírito Santo (ES) and Rio de Janeiro (RJ), while a strong female bias of 94% was observed in the northern regions, Sergipe (SE) and Bahia (BA) (*Marcovaldi et al., 2016*).

Global temperature increases could skew these ratios, impacting adult sex ratios (*Mrosovsky & Provancha, 1992*; *Lolavar & Wyneken, 2015*). Increased male breeding frequency may balance female-skewed hatchling ratios, producing a more equitable operational sex ratio (*Hays et al., 2010*; *Wright et al., 2012*; *Hays, Mazaris & Schofield, 2014*). If males cannot be censused, breeding sex ratio (BSR) can be estimated through paternity testing.

Studies show male reproductive patterns significantly affect long-term population persistence (*Mitchell & Janzen, 2010*; *Hays, Mazaris & Schofield, 2014*). Female sea turtles often mate with multiple males, resulting in nests with multiple paternal contributions (*Lee et al., 2018*). The extent of multiple mating remains uncertain (*Moore & Ball, 2002*), but females can store sperm in the oviduct after mating (*Uller & Olsson, 2008*) for at least 3 months (*Sakaoka et al., 2013*; *Lasala, Hughes & Wyneken, 2020*). Polyandry, as evidenced by multiple paternity (MP), is a common mating strategy observed across several sea turtle species. This strategy has been documented in species such as the green turtle (*Chelonia mydas*) (*Turkozan et al., 2019*), hawksbill (*Eretmochelys imbricata*) (*González-Garza et al., 2015*), leatherback (*Dermochelys coriacea*) (*Stewart & Dutton, 2014*), olive ridley (*Lepidochelys olivacea*) (*Duran et al., 2015*), Kemp's ridley (*Lepidochelys kempii*) (*Kichler et al., 1999*), and flatback (*Natator depressus*) (*Theissinger et al., 2009*).

Multiple paternity in loggerheads has been well-documented, with females typically mating with several males and consequently laying clutches that carry varied paternal contributions. In the United States, specifically in Florida, levels of polyandry range from 22% to 70% (*Bollmer et al., 1999*; *Moore & Ball, 2002*; *Lasala, Hughes & Wyneken, 2018*, *2020*), while in Georgia, this figure reaches 78% (*Lasala et al., 2013*). In Australia, the levels of multiple paternity vary between 48% (*Tedeschi et al., 2015*) to 65.5% (*Howe et al., 2018*). In Turkey, the percentage is 70% (*Sari, Koseler & Kaska, 2017*). Greece, on the other hand, reports the highest level of multiple paternity ever recorded for loggerheads, at 95% (*Zbinden et al., 2007*). In Japan, two studies reported levels of 42.9% (*Sakaoka et al., 2011*) and 27.3% (*Sakaoka et al., 2013*), respectively; however, these studies were conducted on captive turtles. These findings highlight the complex mating system of loggerhead turtles and underscore the importance of incorporating multiple paternity into conservation strategies.

Improving our understanding of multiple paternity and BSR is critical in predicting the long-term persistence of the Brazilian loggerhead populations. Therefore, this study is a crucial step in the conservation of sea turtles. We therefore provide some basic knowledge on the extent of multiple paternity in loggerhead nests on the Atlantic coast in Espírito

Santo, Brazil, together with an estimate of the breeding sex ratio for this population. This research not only fills gaps in our knowledge but also provides a base for future studies that will ensure the survival and genetic health of loggerhead turtles in Brazil and beyond. More specifically, the main objectives of this study were to test for multiple paternity in loggerhead nests along the Atlantic coast in a rookery of Espírito Santo, Brazil, and to estimate the breeding sex ratio for this population. In addition, this study aims to investigate whether there is a relationship between female size and the number of males contributing to the nest. Understanding this correlation could provide insights into how phenotypic traits such as female size may influence reproductive strategies and the multiple paternity.

## MATERIALS AND METHODS

### Ethical statement

This study was performed under the Biodiversity Authorization and Information System license from Chico Mendes Biodiversity Institute (SISBIO/ICMBio), numbers 60690-1 and 60690-2. This study was conducted in accordance with the ARROW guidelines for studies involving wildlife. Institutional approval for animal care and use was obtained, and the relevant application number for study tracking is 07/2019. Additionally, we followed all national, international, and institutional animal care guidelines, including those established by the National Council for the Control of Animal Experimentation (CONCEA). Our research complied with all applicable animal care legislation in Brazil, as set forth by Law No. 11,794/2008, which regulates the scientific use of animals. We took all possible measures to adhere to the 3R principles (Reduction, Refinement, and Replacement), which include minimizing the number of animals used through rigorous planning and optimization of experimental methods (Reduction); ensuring animal welfare throughout the study by constantly monitoring the animals to reduce suffering (Refinement); and whenever possible, replacing the use of tissue from live hatchlings with the residual yolk, aiming to minimize the impact on the animal's organism and reduce the need for direct use of the animal in the procedures (Replacement). Access to Genetic Heritage was registered under SisGen (#A32C980, #A622566, and A2E6361).

### Study area

The sampling of nesting females took place along 13 km of Povoação Beach in the Linhares district, located in the northern part of the state of Espírito Santo, Brazil (Fig. 1). Povoação Beach is part of the extensive 411-kilometer-long coastline of Espírito Santo, which stretches from the border with Bahia to the north and extends southward to the border with Rio de Janeiro, along the South Atlantic Ocean (*Albino, Contti Neto & Oliveira, 2016*). Povoação Beach is characterized as an exposed siliciclastic beach, associated with sandy terraces (*Albino, Contti Neto & Oliveira, 2016*), located in the deltaic plain of the Doce River, where it receives significant riverine inputs (*Dominguez, da Bittencourt & Martin, 1981*) (Fig. 1). This beach, along with Praia de Comboios, is one of the two most important long-term nesting beaches in Espírito Santo and is crucial for sea turtle conservation efforts (*Marcovaldi & Chaloupka, 2007*).
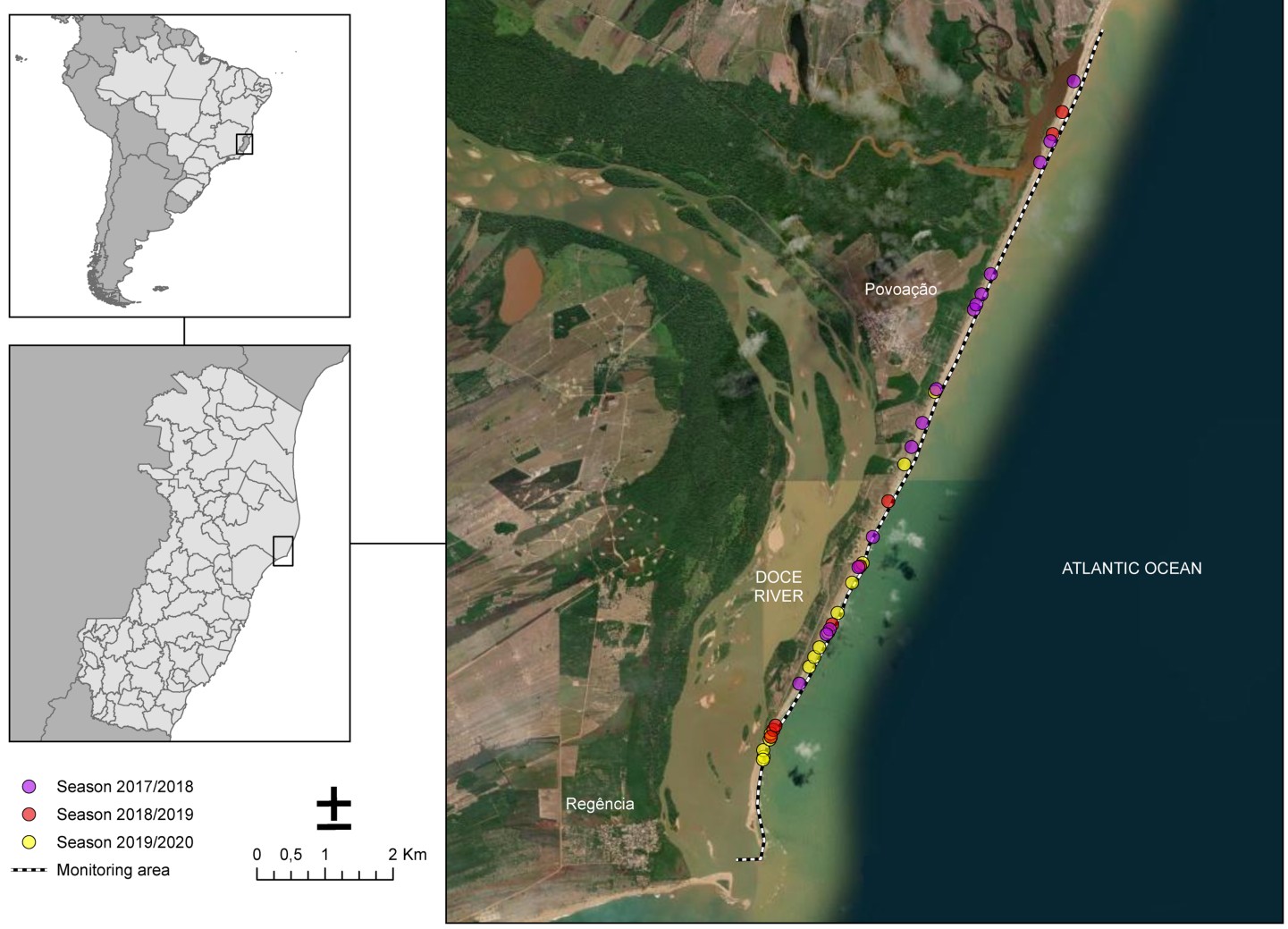

**Figure 1 Study area along the coast near the mouth of the Doce River showing the monitoring sites for the 2017/2018, 2018/2019, and 2019/2020 seasons.** Nesting beach at Povoação, located within Espírito Santo state, Brazil, where the study was conducted. The dotted line represents the monitored area along the beach. The colored dots indicate the locations of loggerhead turtle nests observed across different breeding seasons: pink for the 2017/2018 season, red for the 2018/2019 season, and yellow for the 2019/2020 season.

## Sampling procedures

Samples were collected from loggerhead sea turtles over three consecutive nesting seasons (2017/18, 2018/19, and 2019/20) under License numbers 60690-1 and 60690-2 (SISBIO/ICMBio).

Sampling occurred in the nesting seasons between October and December. During night monitoring, females were observed emerging from the ocean to nest. Once oviposition was completed and the turtle began covering her nest, tissue samples were collected from the proximal region of the female's anterior flipper using 6 mm punches (Acu-Punch®, North Sydney, NSW, Australia) and stored in tubes with 95% ethanol solution. Flipper tags were applied following standard procedures (*Santos et al., 2013*) to ensure that each turtle was sampled only once. Global Positioning System points were

taken for the location of each nest. Each nest was marked with numbered stakes, and a protective screen was placed to prevent predation and to reduce the likelihood of invasion by other turtles. A diameter plastic net was attached to the metal screen to retain hatchlings upon emergence. After 45 days of incubation, the nests were checked daily for hatchling emergence, with collection occurring after 5 AM. Up to 20 hatchlings were genotyped from each nest, following the guidelines suggested by *DeWoody et al. (2000)*, which indicate that this sample size is sufficient to detect all parental contributions in a half-sib progeny array. Hatchlings were sampled randomly to ensure no bias related to their position in the nest or emergence timing. Live hatchlings were sampled using buccal swabs or residual yolk. In cases where the net used to retain hatchlings was ineffective, resulting in the loss of hatchlings from several nests, tissue from dead hatchlings or embryos found in the nest was sampled instead. All tissue samples were stored in labeled tubes containing 95% ethanol. This sample size is sufficient to detect all parental contributions in a half-sib progeny array (*DeWoody et al., 2000*). It is important to note that if multiple paternity was not detected in the sample of 20 hatchlings, it is statistically unlikely to be detected in the remaining hatchlings from that nest (*Lasala, Hughes & Wyneken, 2020*). Hatchling samples were excluded if two or more loci did not amplify or if maternal contributions could not be verified (lack of amplification).

## DNA isolation

We isolated the genomic DNA (gDNA) from all samples using a DNA salt extraction protocol (SDS/NaCl/Proteinase K) following the instructions of *Bruford et al. (1992)*. All gDNA samples were evaluated through electrophoresis in 1% agarose gel stained with Blue Green® dye (LGC Biotechnology, Teddington, England) and visualized on the UV-transilluminator L-PIX Touch 20 × 20 cm (Loccus©, Jaipur, Rajasthan). The gDNA samples were quantified using a NanoDropND-100 spectrophotometer (Thermo Fisher Scientific, Waltham, MA, USA) and diluted to 50 ng/μl.

## Microsatellites genotyping

Five nuclear microsatellites were initially selected for this study: Cc7G11, Cc1F01, Cc1G02, Cc1G03, and CcP7D04 (*Shamblin et al., 2007*, *2009*). However, the microsatellite Cc7G11 did not consistently amplify in the hatchlings and was consequently excluded from the paternity analyses. The remaining four microsatellite loci were successfully amplified using polymerase chain reactions (PCR) in 13.5 μl mixes containing Buffer 1× (Invitrogen®, Waltham, MA, USA), 1.5 mM MgCl2 (Invitrogen®, Waltham, MA, USA), 0.2 mM dNTPs, 0.16 μM of each primer (forward and reverse), 0.16 μM of fluorescence labelled M13 tag following the protocol proposed by *Schuelke (2000)* (6-FAM, PET, NED, or VIC), 0.5 U of Platinum Taq (Invitrogen®, Waltham, MA, USA), and 1 μl of DNA (50 ng/μl).

For the markers Cc1F01, Cc1G02, and Cc1G03, PCRs were performed using a VeritiTM thermocycler (Applied Biosystems, Waltham, MA, USA) with the following cycle: initial denaturation at 94 °C for 3 min, followed by 30 cycles of 94 °C for 30 s, 57 °C for 30 s, and 72 °C for 30 s. Additional eight cycles of 94 °C for 30 s, 53 °C for 30 s, and 72 °C for 30 s were used to anneal the fluorescence M13 tag to the fragments, with a final extension at

72 °C for 30 min (*Schuelke, 2000*). For the CcP7D04 marker, the conditions of *Schuelke (2000)* were adapted to a PCR touchdown protocol to optimize amplification and minimize non-specific bands: initial denaturation at 95 °C for 5 min, followed by 20 cycles of 95 °C for 20 s, 60 °C for 20 s (decreasing 0.5 °C per cycle), and 72 °C for 30 s. This was followed by 20 more cycles of 95 °C for 20 s, 50 °C for 30 s, and 72 °C for 30 s, with a final extension at 72 °C for 10 min. Positive and negative controls were included in all PCRs to check for potential contamination. PCR products were confirmed through electrophoresis in a 1% agarose gel stained with Blue Green® dye (LGC Biotechnology, Teddington, England), using a 100 bp Ladder (Ludwig Biotec©, Alvorada, Brazil), and visualized on a UV-transilluminator (L-PIX Touch 20 × 20 cm; Loccus©, Jaipur, Rajasthan). The positive products were then multiplexed in a mix containing 7.0 µl of formamide (Hi-Di Formamide; Applied Biosystems, Waltham, MA, USA), 0.5 µl of GeneScan™600 LIZ™ fluorescent molecular size standard (Applied Biosystems, Waltham, MA, USA), and 0.5 µl of PCR products from each of the four microsatellite loci, followed by genotyping on an ABI 3500 Automatic DNA Analyzer (Thermo Fisher Scientific, Waltham, MA, USA). All molecular procedures were conducted at the Núcleo de Biodiversidade Genética Luiz Paulo de Souza Pinto (NuBiGen, https://nubigen.ufes.br/) at the Universidade Federal do Espírito Santo (UFES), Brazil. A subset of samples (10%) was re-run in the same PCR reaction to assess the genotyping error rate.

Although the number of microsatellite loci used in this study are fewer than in some research (*e.g.*, seven in Sanibel Island, Florida: *Lasala, Hughes & Wyneken, 2018*, *2020*) other studies have used less (*e.g.*, two in Melbourne Beach, Florida: *Bollmer et al., 1999*; and two in Turkey: *Sari, Koseler & Kaska, 2017*). Four microsatellite loci are comparable to several other works, including those from Melbourne Beach (Florida) (*Moore & Ball, 2002*), Australia (*Tedeschi et al., 2015*), and Greece (*Zbinden et al., 2007*). Our selection was guided by the quality and consistency of amplification and the imperative to minimize genotyping errors. Despite the smaller number, the four loci used are highly polymorphic and have been validated in previous research (*Shamblin et al., 2007*, *2009*). Moreover, the chosen markers demonstrate excellent values of Probability of Identity (PI) and Probability of Exclusion (PE), further underscoring their effectiveness. These attributes make the selected loci not only adequate in number but also highly effective for genetic studies of sea turtles.

### Data analysis

Alleles were manually scored and standardized following the instructions of *Shamblin et al. (2007*, *2009)*, with the mothers' and hatchlings' genotypes established using the software Geneious R9 (*Kearse et al., 2012*). The observed (Ho) and expected heterozygosity (He), deviations from Hardy-Weinberg Equilibrium (HWE) with Bonferroni correction, and presence of null alleles were obtained through the PopGenReport package for R (*Gruber & Adamack, 2015*; *RStudio Team, 2024*).

## Paternity analysis

We determined the PI and the PE using a larger female dataset ($N = 236$) from the same population, available in *Ludwig et al. (2023)*, with the software GenAlEx (*Peakall & Smouse, 2012*). The PI provides the likelihood that two unrelated random samples will have the exact same genotype based on the estimated allelic frequencies of the mothers, while the PE provides the proportion of the population with a genotype containing at least one allele not present in the mixed profile (when only one parent is known). These measures help to validate the effectiveness of the genetic markers in accurately identifying and distinguishing individuals within the population and in ensuring the accuracy of parentage assignments.

The hatchlings' genotypes were compared to their mother's genotypes to exclude shared alleles and determine the minimum paternal contribution to each nest (*Lasala, Hughes & Wyneken, 2018*). Multiple paternity was considered when hatchlings from the same nest shared three or more paternal alleles from two or more microsatellite loci (*Yue & Chang, 2010*). A paternity analysis was performed using the exclusion method with the software COLONY 2.0 (*Jones & Wang, 2010*), which implements a maximum likelihood method to suggest the maximum number of potential fathers for each nest based on the available genotypes. The COLONY parameters included a polygamous mating system for both sexes in a diploid species. A medium run was conducted with intermediate results monitored every 1 s. The probability of a mother being included in the female candidates was set to 1. The breeding sex ratio was calculated by dividing the overall number of males identified by the overall number of females used in the study.

To examine the relationship between sample size (number of genotyped hatchlings) and the frequency of multiple paternity, we conducted a Pearson's correlation analysis ($\rho$). This analysis assessed the correlation between the number of genotyped hatchlings per nest and the number of inferred males (as an indicator of multiple paternity). Statistical significance was evaluated at a 95% confidence level.

## Female size *vs.* frequency of multiple paternity analysis

Measurements were taken from all undamaged females. The measurements included the curved carapace length (CCL) and the curved carapace width (CCW), using a measuring tape with 1.0 mm precision (Table S1). The carapace area (CA) was estimated using the formula for the area of an ellipse ($A = \pi \cdot CCL \cdot CCW$), assuming an approximately elliptical shape.

To examine the relationship between female size and the frequency of multiple paternity, we used a generalized linear model (GLM) with a Poisson distribution and a log link function to further investigate the relationship between female carapace size and the number of fathers per nest, accounting for the number of genotyped hatchlings. The predictors included CCL, CCW, and CA (modeled separately), with the number of genotyped hatchlings included as an offset term to control for the variability in sample size across nests. This approach ensured that nests with fewer genotyped hatchlings did not disproportionately affect the results. All analyses were conducted in RStudio version 2024.04.2+764 (*RStudio Team, 2024*).

## Maps

The maps were generated using QGIS 3.34 (*QGIS Association, 2024*). The cartographic base for the South American continent, Brazil, and Espírito Santo originates from the Integrated System of Geospatial Bases for the state of Espírito Santo—GEOBASES, accessed through a Web Feature Service (WFS) connection.

## RESULTS

A total of 66 females corresponding to 69 natural nests were monitored. Due to storms and predation during the incubation period, many nests were lost. A total of 1,109 hatchlings from 43 nests attributed to 42 females were initially analyzed. These nests included 17 from the 2017/18 season, 11 from the 2018/19 season, and 15 from the 2019/20 season (Fig. 2; for more details, see Table S1). However, only 534 hatchlings (48.15%) were successfully genotyped, as some samples, particularly those from residual yolks, were degraded and did not yield reliable results (Fig. 2; Table S1). Despite nearly half of the hatchlings being successfully genotyped, the number of nests and females per season remained consistent. Notably, female SMV141 nested in two different reproductive seasons (2017/18 and 2019/20), which accounts for 42 females but 43 nests. The number of hatchlings analyzed per nest ranged from 3 to 20, with an average of 12.41 hatchlings per nest.

The number of alleles per locus ranged from 10 to 13 for nesting females ($N = 42$) (Table 1). There was no evidence of deviation from the Hardy-Weinberg equilibrium for any loci after the Bonferroni correction. The expected heterozygosity (He) by locus for the female dataset ranged from 0.781 to 0.848, while the observed heterozygosity (Ho) by locus ranged from 0.818 to 0.932 (Table 1). There was no evidence of null alleles. The PI using the four loci was $3.2 \times 10^{-6}$, and the PE found was 99.10%. The genotyping error rate was zero for the four loci used in this study.

In total, 88 distinct males were identified contributing to the hatchlings across all seasons, with 38 males in 2017/18, 24 in 2018/19, and 26 in 2019/20 (Figs. 2 and 3; Table S2). The total number of males does not represent a simple sum since some males contributed to multiple nests, either within the same season or across different seasons. For example, ♂3 contributed to both the 2017/18 and 2019/20 seasons (Figs. 2 and 3; Table S2). Polyandry, or multiple paternity, was observed in 31 out of 43 nests (72.09%), with the number of males per nest ranging from one to six, and an average of 2.04 males per nest. The Pearson's correlation analysis between the number of genotyped hatchlings per nest and the number of inferred males revealed a significant negative correlation ($p$-value = 0.007 and $\rho = -0.401$).

Twelve males were observed to contribute to more than one nest, either within a single season or across multiple seasons (Figs. 2 and 3; Table S2). Among these, seven males displayed polygyny by fertilizing multiple females within the same season. Specifically, six of these polygynous males were active during the 2017/18 season (♂1, ♂3, ♂8, ♂21, ♂22, and ♂28), while one male was from the 2018/19 season (♂25). These males contributed to as many as three different nests within the same season (Fig. 3; Table S2). Moreover, some of these males also contributed across different seasons. For example, ♂3, ♂8, and ♂21 fertilized two nests in the 2017/18 season and another in the 2019/20 season; ♂7, ♂11 and

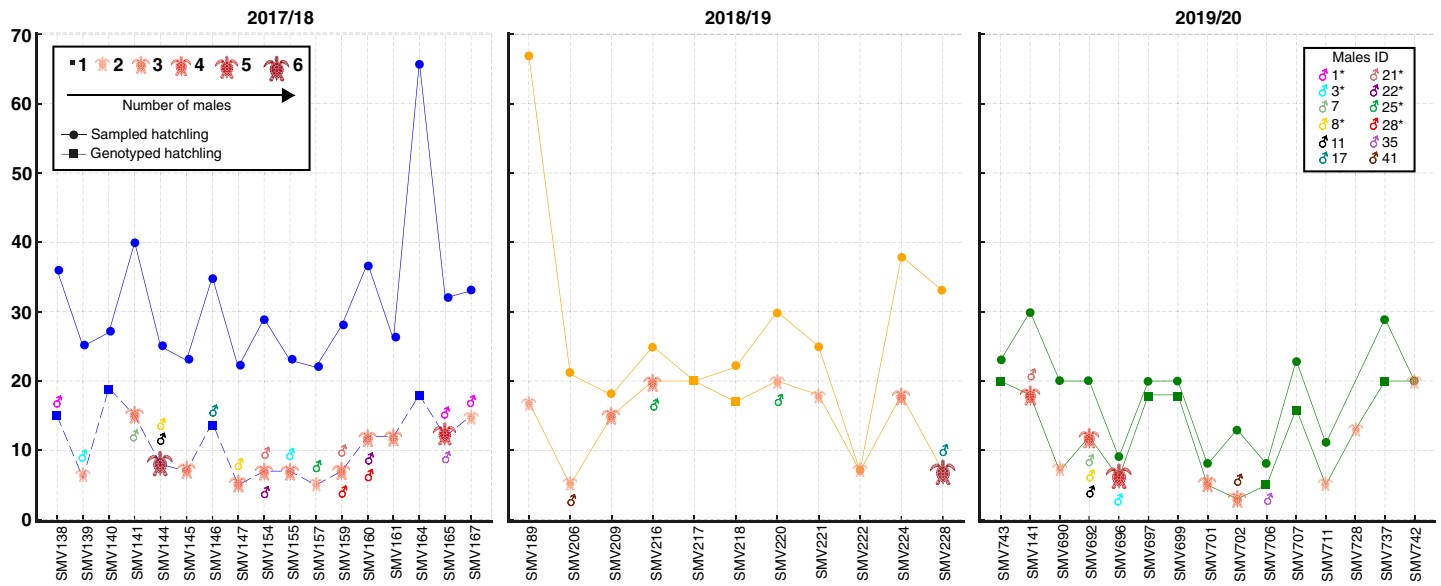

**Figure 2 Distribution of sampled and genotyped hatchlings per nesting site across the 2017/2018, 2018/2019, and 2019/2020 seasons, with identified contributing males.** Analysis of loggerhead turtle nests across three nesting seasons: 2017/2018 (blue), 2018/2019 (yellow), and 2019/2020 (green). The x-axis lists individual nests analyzed per season, while the y-axis indicates the number of sampled hatchlings (circles) and genotyped hatchlings (squares). The size and color of the turtle icons reflects the number of males contributing to each nest. Males exhibiting polygyny are denoted by an asterisk (*) next to their ID in the legend.

**Table 1 Genetic loci statistics for loggerhead females at Povoação Beach, Brazil (NA, LR, Dye, He, Ho, pHWE).**

| Locus | $N_A$ | LR (Dye) | He | Ho | pHWE |
|-------|-------|----------|------|------|------|
| CcP7D04 | 11 | 343–399 (NED) | 0.781 | 0.833 | 1 |
| Cc1F01 | 10 | 304–372 (VIC) | 0.815 | 0.818 | 0.25 |
| Cc1G02 | 11 | 260–308 (NED) | 0.848 | 0.932 | 1 |
| Cc1G03 | 13 | 277–333 (6-FAM) | 0.781 | 0.833 | 1 |

**Note:**
Descriptive statistics for each locus from Povoação Beach, Brazil. Data from loggerhead nesting females only ($n$ = 42). NA, number of alleles detected in this population; LR, locus range which accounts for where alleles were found; Dye, the fluorescent tag associated with each locus; He, expected heterozygosity; Ho, observed heterozygosity; pHWE: $p$-value for Hardy-Weinberg equilibrium.

♂35 fertilized one nest in 2017/18 and another in 2019/20; ♂17 fertilized one nest in 2017/18 and another in 2018/19; ♂25 fertilized one nest in 2017/18 and two in 2018/19; and ♂41 fertilized one nest in 2018/19 and another in 2019/20.

Among the twelve males that fertilized more than one nest, five (♂7, ♂11, ♂17, ♂35, and ♂41) were involved in nests across multiple seasons without displaying polygyny (Figs. 2 and 3; Table S2). The polygyny rate, calculated from the total of 88 males and the seven events of polygyny, was 7.95%.

Furthermore, eight males (20.5%) from the 2017/18 season continued to contribute to fertilization in subsequent seasons. Two males returned in the 2018/19 season (♂17 and ♂25), while six males returned in the 2019/20 season (♂3, ♂7, ♂8, ♂11, ♂21 and ♂35) (Figs. 2 and 3; Table S2). This pattern suggests a degree of site fidelity and indicates a stable

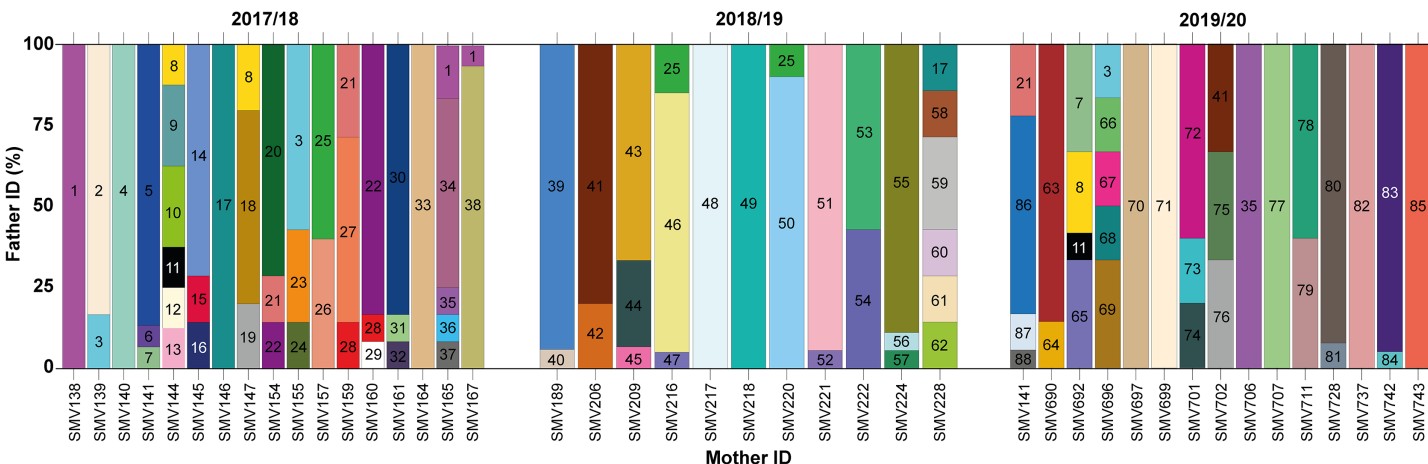

**Figure 3 Proportion of paternity contributions from identified males to hatchlings across different females during the 2017/2018, 2018/2019, and 2019/2020 nesting seasons.** Paternal contributions to loggerhead turtle nests across three nesting seasons: 2017/2018, 2018/2019, and 2019/2020. Each vertical bar on the x-axis corresponds to an individual nest, identified by the mother's ID. The y-axis represents the percentage of contribution from different males to each nest. Each color block within a bar represents a different male contributing to that particular nest.

breeding population within the aggregation. The overall breeding sex ratio for this nesting aggregation was approximately 1 female for every 2.09 males, or 42 females to 88 males.

In examining the relationship between female size and the number of males, the data for CCL shows a slight upward trend as the number of males increases (Fig. 4). However, there is significant data variability, particularly in the groups with two and three males. The statistical analysis reveals a $p$-value of 0.716 and a $\rho$ of 0.058, indicating that this trend is not statistically significant, suggesting that the apparent increase may be due to random variation rather than a true underlying effect.

Similarly, the CCW data exhibit no clear trend in relation to the number of males (Fig. 4). The distribution of the data points does not indicate a consistent pattern, with the $p$-value of 0.622 and $\rho$ of 0.077 further supporting the lack of a significant correlation. This suggests that the width of the carapace does not meaningfully change with varying numbers of males.

The CA shows only modest variation across different numbers of males, with a slight increase observed in groups with five and six males (Fig. 4). However, this variation is not substantial, as indicated by the $p$-value of 0.645 and $\rho$ of 0.074, which suggest that there is no significant correlation. The data imply that, much like with CCL and CCW, any observed differences in the number of males are likely due to random fluctuations rather than a direct effect of the size of females.

The results of the GLM analysis for CCL and CCW did not reveal a significant relationship between these size metrics and the number of males per nest. The GLM for CCL yielded an estimate of 0.589, Standard Error (SE) = 2.768, and $p$-value = 0.831, and for CCW, an estimate of 1.020, SE = 3.2444, $p$-value = 0.753. These results further confirm the lack of significant influence of these metrics on the number of contributing males. Similarly, the GLM analysis for CA showed no significant effect on the number of males, with an estimate of 1.373, SE = 1.429 and $p$-value = 0.3369. Although the intercept for CA

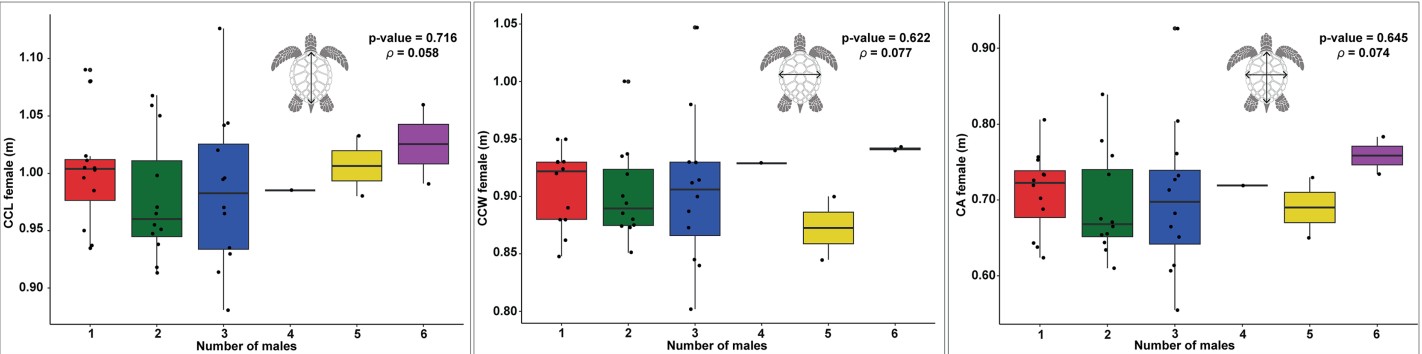

**Figure 4 Boxplots showing the relationship between the number of males and female body size measurements (CCL, CCW, and CA) across different nesting seasons, with statistical analysis of correlations.** Relationship between female carapace size and the number of males contributing to loggerhead turtle nests, as measured by curved carapace length (CCL), curved carapace width (CCW), and carapace area (CA). The colors represent the different numbers of males contributing to each nest.

was significant (Intercept = −2.598, SE = 1.016, $p$-value = 0.0106), this result suggests that carapace area does not meaningfully explain the variation in the number of fathers per nest.

## DISCUSSION

This study is the first to assess multiple paternity in loggerhead turtles in Brazil using nuclear DNA markers. Additionally, this is the first study to report instances of polygyny in loggerhead turtles. Similar high rates of polygyny have been observed in other sea turtle species, such as hawksbill turtles in El Salvador, where 31.8% of the males sired nests from multiple females (*Gaos et al., 2018*), and in Mediterranean green turtles in Turkey, with a polygyny rate of 4.4% (*Turkozan et al., 2019*). Factors that may facilitate polygyny include a more balanced primary sex ratio (*Marcovaldi et al., 2016*) and reproductive philopatry (*FitzSimmons et al., 1997*). In our study, analyzing three nesting seasons provided a broader understanding of polygyny dynamics by accounting for annual variability in the population.

In highly endangered species, the difficulty in finding mates due to a low number of males could lead to an increase in polygyny events (*Crim et al., 2002*). This behavior may also suggest that mating occurs closer to nesting beaches, particularly when the species' range is limited (*Natoli et al., 2017*). Some studies propose that polygyny might be a consequence of a female-biased sex ratio (*Crim et al., 2002*). However, it remains unclear whether polygynous mating strategies can effectively counterbalance the potential impacts of feminization and the reduced availability of males (*Gaos et al., 2018*). Further research is needed to determine how these mating systems will adapt to ongoing environmental changes and what implications this will have for the conservation of these iconic marine species. These findings can also benefit captive breeding programs by helping to maintain genetic diversity and improve breeding success in controlled environments (*Maggeni & Feeney, 2020*).

Historically, research on sea turtles has predominantly concentrated on studying females, particularly during their time on the beach, resulting in a significant gap in our

understanding of males (*Bjorndal, 1999*). This bias in research has left male's behavior largely underexplored. When studies on male turtles do occur, they are seldom conducted through systematic, active searches. Instead, they tend to rely on incidental captures or data collected from stranded individuals (*Casale et al., 2005*; *James, Eckert & Myers, 2005*; *Kawazu et al., 2014*). While these methods can yield valuable insights, they are inherently limited by their reliance on chance encounters and the logistical constraints that make them impractical for many researchers. While informative, these methods are not feasible for many researchers and often rely on random chance. In contrast, the method of multiple paternity, adopted here to assess successful male contributions, is both effective and informative, offering a more reliable approach to studying male sea turtles.

The PI ($3.2 \times 10^{-6}$) and the PE (99.10%) obtained in this study are comparable to previously published studies on loggerhead populations and are consistent with the proposed reliability values for these indices in other taxa, as emphasized by *Waits, Luikart & Taberlet (2001)*. Three recent articles in North America, reported similar PI and PE values: Georgia—$5 \times 10^{-13}$ and 99.9%, respectively; SW Florida $7.3 \times 10^{-13}$ and 100%; and NW Florida: $<9 \times 10^{-8}$ (*Lasala et al., 2013*; *Lasala, Hughes & Wyneken, 2018*; *Silver-Gorges et al., 2024*). These three studies used between 5–16 microsatellites, but still had similar probability results. The power of these analyses depend on the number of alleles per locus and their frequency distribution in the population. A higher number of alleles and a balanced distribution increase the ability to distinguish between individuals, which in turn improves the PI and PE values. This suggests that the genetic markers used in this study, despite being fewer in number (four microsatellites), provide a level of resolution that is sufficient for distinguishing individual genotypes within the population (Table 1). Our high PE value further supports the robustness of our methodology, indicating a high probability of correctly excluding non-parental individuals from paternity, thereby reducing the likelihood of false positives in identifying paternity events.

The reliable detection of multiple paternity with only four microsatellites is particularly valuable for studies with limited samples, enabling more cost-effective studies of genetic diversity and reproductive behavior in other loggerhead populations and, potentially, in other species where similar challenges exist. While future research could explore the integration of additional markers or next-generation sequencing techniques to further enhance the resolution and accuracy of paternity analysis, this study provides a strong foundation for using the minimum number of microsatellites in genetic monitoring and conservation efforts.

Loggerhead turtles in Espírito Santo may exhibit higher levels of promiscuity compared to other breeding populations globally, potentially influenced by specific characteristics of the Brazilian population. This conclusion is supported by the unique mating behaviors observed, including the first documentation of polygyny in this species, suggesting a breeding strategy that differs from those seen elsewhere. This distinct pattern of mating behavior could enhance genetic diversity and reproductive success, highlighting the potential for increased promiscuity among both sexes in this population.

Alongside these observations on mating behavior, we also examined the influence of the number of genotyped hatchlings on the detection of multiple paternity. Many nests with

eight to 15 genotyped hatchlings showed contributions from two to three males. This detection pattern may result from random sampling of smaller nests, or it could be that nests with lower reproductive success are hiding the true number of male contributions. While significant, the variability in the data suggests that sperm competition or fertilization success, may also play a role. Further research is needed to understand the biological and ecological drivers of this pattern.

Sperm storage, which allows females to ensure fertilization of nests throughout the nesting season (*Uller & Olsson, 2008*; *Sakaoka et al., 2013*; *Lasala, Hughes & Wyneken, 2020*), further contributes to this reproductive flexibility. While females can be selective in their mate choice (*Kokko & Mappes, 2013*), the observed contributions from multiple males across various females' nests raise intriguing questions. This could suggest that either (a) there are fewer males available in this population, limiting female choice, or (b) this behavior represents a distinctive reproductive strategy unique to the region, reflecting adaptations to local ecological conditions.

A study analyzing mtDNA haplotypes revealed that the nesting aggregation of loggerhead turtles in Espírito Santo is genetically distinct from other rookeries in Brazil (*Shamblin et al., 2014*). The high rate of multiple paternity observed in this region may be attributed to increased male breeding periodicity (*Hays, Mazaris & Schofield, 2014*). When males and females converge in the same breeding area, the density of individuals can be up to a hundred times greater than when they move independently, potentially linking the incidence of multiple paternity to turtle density (*Lee et al., 2018*). This elevated occurrence of multiple paternity suggests that such behavior may not confer a clear evolutionary advantage but would rather be a byproduct of increased encounters between males and females. When comparing this with other loggerhead rookeries, we observe variable rates of multiple paternity. For example, Greece has a high rate of 93.3% (*Zbinden et al., 2007*), likely due to the restricted movement and high density of turtles in that area (*Lee et al., 2018*). In contrast, rookeries in Florida, such as Melbourne Beach, show a much lower rate of 31% (*Moore & Ball, 2002*), possibly due to larger foraging areas and lower densities. Similarly, Wassaw Island, Georgia, has a higher multiple paternity rate of 75% (*Lasala et al., 2013*), reflecting intermediate density conditions and movement patterns. Additionally, more monogamous females could be influenced by phenotypic traits or the timing of nesting, as females nesting early or late in the season may have fewer mating opportunities.

Moreover, rising evidence indicates a female-skewed hatchling sex ratio in sea turtle nesting populations (*Hays, Mazaris & Schofield, 2014*). In Espírito Santo, the estimated hatchling sex ratio was 53% females to 47% males (*Marcovaldi et al., 2016*), which is much closer to a balanced 50/50 ratio compared to other Brazilian nesting populations. For instance, the northern nesting aggregation, including Sergipe and Bahia, reported a ratio of 94% females (*Marcovaldi et al., 2016*). Interestingly, both green and loggerhead turtles show a high incidence of multiple paternity even in environments with biased hatchling sex ratios (*e.g.*, 80% females) (*Hays et al., 2023*).

In summary, this suggests that operational sex ratios could be more balanced due to specific male mating behaviors (*Hays, Mazaris & Schofield, 2014*). Some studies propose

that male behavior, such as frequent breeding, visiting multiple rookeries (*Wright et al., 2012*), and engaging in polygyny (*Bell et al., 2010*) can sustain sea turtle fertility even at low population sizes. The breeding sex ratio (BSR) for Espírito Santo found in this study, 2.09 males per female, aligns with other population genetic data. Loggerhead population structure is driven by female natal nesting fidelity, coupled by male mediated gene flow (*Bowen et al., 2005*). This BSR is consistent with regions that display female-biased primary sex ratios, which may be compensated by more male-biased operational sex ratios (*Hays et al., 2010*). These findings support the notion that the unique mating dynamics and genetic characteristics of loggerhead turtles in Espírito Santo are shaped by local ecological conditions, further underscoring the complexity of reproductive strategies in this population.

Studies on the reproductive behavior of sea turtles have demonstrated that polyandry is a common mating strategy (*Wright et al., 2012*; *Lasala, Hughes & Wyneken, 2018*; *Lee et al., 2018*). The multiple paternity rate of 72.09% found in this study aligns well with global patterns observed in other research (Fig. 5; Table S3) (*Bollmer et al., 1999*; *Moore & Ball, 2002*; *Zbinden et al., 2007*; *Lasala et al., 2013*; *Tedeschi et al., 2015*; *Sari, Koseler & Kaska, 2017*; *Howe et al., 2018*; *Lasala, Hughes & Wyneken, 2018, 2020*). Figure 5 shows significant variation in multiple paternity rates across different geographic locations, ranging from 15.8% in northwest Florida (*Silver-Gorges et al., 2024*) to 95% in Greece (*Zbinden et al., 2007*). Additionally, the average number of fathers per nest also varies, with values from 1.40 in Florida (*Moore & Ball, 2002*) to 3.50 in Greece (*Zbinden et al., 2007*) (Fig. 5; Table S3), further highlighting the diverse reproductive strategies among loggerhead turtles worldwide. The maximum number of fathers per nest observed in this study was six. This is comparable to other sea turtle species, such as leatherback, with up to three fathers (*Figgener et al., 2016*), green turtles, where up to ten fathers have been reported (*Turkozan et al., 2019*), hawksbill turtles, with up to three fathers (*González-Garza et al., 2015*), Kemp's ridley turtles, with three fathers (*Kichler et al., 1999*), olive ridley turtles, with up to four fathers (*Jensen et al., 2006*), and flatback turtles, also with up to four fathers (*Theissinger et al., 2009*).

The variability in mating behavior may be attributed to the loggerhead's extensive migratory distribution (*Dodd, 1988*), which facilitates opportunistic mating between feeding areas and migratory corridors (*Lasala, Hughes & Wyneken, 2018*). As global temperatures continue to rise, leading to skewed sex ratios among hatchling and juvenile populations (*Jensen et al., 2018*), the prevalence of multiple paternity in sea turtle clutches suggests that adult male turtles are not constrained in these breeding populations (*Hays, Shimada & Schofield, 2022*). Understanding these patterns is crucial for developing effective conservation strategies, especially as climate change continues to impact sea turtle populations worldwide. Future studies should aim to explore the implications of these reproductive strategies on genetic diversity and population resilience in the face of environmental changes.

For female sea turtles, multiple mating offers significant evolutionary advantages, such as increasing the likelihood of successful fertilization and enhancing the genetic diversity of their hatchlings (*Andersson, 1994*; *Pearse & Avise, 2001*; *Calsbeek et al., 2007*). Increased

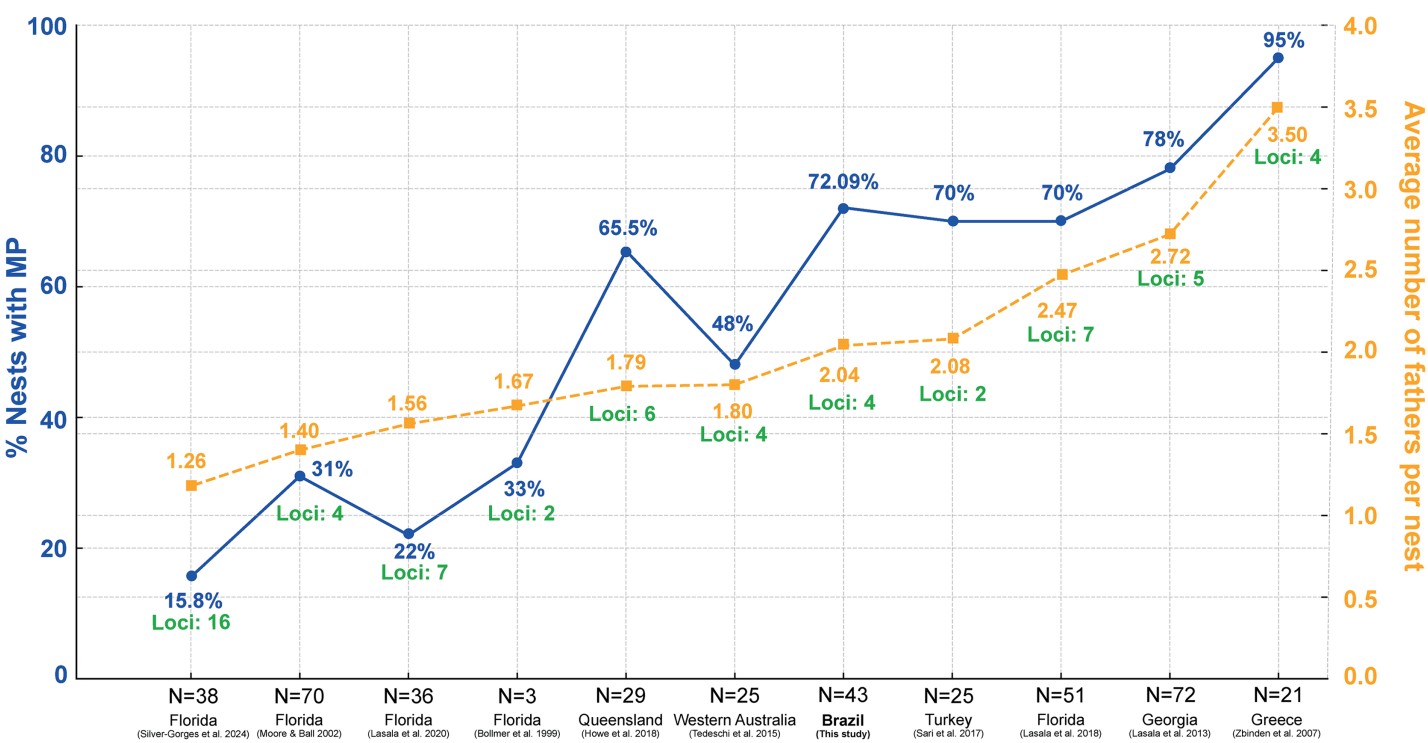

**Figure 5 Comparison of the percentage of nests with multiple paternity (MP) and the average number of fathers per nest across various regions and studies, including loci used in paternity analysis.** Percentage of loggerhead turtle nests with multiple paternity (MP) (blue) across various studies and locations, along with the average number of fathers per nest (yellow) and number of loci used in the analysis (green). N = number of clutches.                                         

genetic diversity is especially crucial in small or recovering populations (*Balazs & Chaloupka, 2004*). A broad gene pool can bolster the population's resilience against evolutionary changes and reduce the risk of inbreeding, which can lead to an extinction vortex (*Frankham, Ballou & Briscoe, 2010*). In the context of a warming climate, maintaining and enhancing genetic diversity through multiple paternity becomes an important evolutionary strategy.

For instance, for a small nesting population in Turkey, the high frequency of MP might be driven by population density, which increases the chances of females encountering multiple males and thus the likelihood of multiple mating events (*Sari, Koseler & Kaska, 2017*). However, the underlying causes of MP are complex and can vary significantly depending on geographic location, environmental conditions, and individual characteristics of the turtles.

The complexity of mating behaviors in sea turtles is evident in the way females may engage in additional mating as a strategy to avoid the high costs associated with resisting male attempts (*Lee et al., 2018*); however, in some species, females actively choose their mates (*Lee & Hays, 2004*). In Georgia, larger females significantly had more successful male contributions (*Lasala et al., 2013*), but in Florida, larger, presumably older, females tended to exhibit greater mate choice, resulting in a lower incidence of MP compared to younger, less experienced females (*Lasala, Hughes & Wyneken, 2020*). Further, in Turkey,

there were no significant connections between female size and MP frequency found (*Sari, Koseler & Kaska, 2017*). Female mate choice may matter by nesting population. Moreover, the assumption that larger females are necessarily older is not fully supported. *Phillips et al. (2021)* demonstrated that sea turtles grow very slowly after reaching maturity, meaning size is not always a reliable indicator of age. Larger females likely matured close to their current size, with factors like juvenile growth rates and resource availability being key influences. Therefore, the relationship between size, age, and experience should be reconsidered in discussions of mate choice and MP in sea turtles. Additionally, cryptic female choice (CFC) could be another important mechanism influencing mating outcomes. CFC refers to a female's ability to influence paternity after mating by selectively using or discarding sperm through physiological or biochemical processes (*Firman et al., 2017*). This mechanism may allow females to exert postmating control over mate selection, potentially explaining why some females show reduced rates of multiple paternity despite engaging in multiple matings.

The data presented by *Sari, Koseler & Kaska (2017)*, along with our results (Fig. 4), indicate a high degree of multiple paternity regardless of female size. This finding supports the idea that factors such as population density and local environmental conditions may play a more significant role in determining the incidence of MP than female size alone.

The GLM analysis presented here further supports these findings. None of the size metrics showed a significant relationship with the number of males contributing to the nests. Although CA was significant, this does not suggest that carapace size fully explains the variation in the number of fathers per nest. These findings confirm that female body size is unlikely to be a key factor influencing multiple paternity in *C. caretta*. Instead, our data suggest that MP in *C. caretta* may be driven by a complex interplay of factors, including female experience, population dynamics, and potentially even the mating strategies employed by the males.

Despite concerns that climate change, particularly the warming of nesting beaches, could lead to a reduction in the number of male sea turtles, thereby limiting female mating opportunities, this may not necessarily lead to decreased genetic diversity. Males are likely capable of breeding annually, and evidence from this study suggests that males can mate with multiple females within a single breeding season. This behavior indicates that MP in sea turtles can be understood as a product of the interaction between individual behaviors, such as mate choice and mating frequency, and broader population characteristics, like density and the availability of males. These factors together help maintain a balanced BSR, even in the face of environmental challenges.

## CONCLUSIONS

This research establishes a foundation for future comparative studies on the multiple paternity of loggerhead turtles in Brazil. While multiple paternity is well-documented in sea turtles, this study breaks new ground by presenting the first global evidence of polygyny in loggerheads and the first record of polyandry in this species within Brazil. Our findings highlight that some males contribute to multiple nests within and across breeding seasons, offering valuable insights into male reproductive strategies.

Future studies in Brazil should build on this work by using the same genetic markers to compare male genotypes across different nesting populations, such as those in Rio de Janeiro, Bahia, and Sergipe. This approach could clarify male gene flow between seasons and regions and provide deeper insights into reproductive population sex ratios. Expanding this research to other Brazilian sub-populations will also help estimate adult sex ratios and assess the impact of female-biased hatchling production on the Brazilian coast.

Given the significant role of multiple mating in influencing effective population size and genetic diversity, continued research on this phenomenon is crucial. Such studies will be vital for developing informed management and conservation strategies, particularly in the context of future global warming scenarios.

## ACKNOWLEDGEMENTS

The authors are grateful to the subject editor and the anonymous reviewers who helped us to improve the quality of this contribution. We thank UFES, NuBiGen and LGEM for the laboratory facilities. The authors thank Juliana Justino, Matheus de Oliveira Fernandes Adão, and Tarsila Mariano Breciani for their support of our research. We are grateful to Projeto Tamar Foundation for assistance in collecting samples, Tamar-ICMBIO Center for providing flipper tags, and Gustavo Stahelin for his valuable comments on the manuscript.

### Funding

This work was supported by Fundação de Amparo à Pesquisa e Inovação do Espírito Santo (FAPES, grant #036/2019), Conselho Nacional de Desenvolvimento Científico e Tecnológico (CNPq, grant #80600417/17), and the Renova Foundation—Brazil through its Technical-Scientific Cooperation Agreement n° 30/2018 with FEST—Brazil. Laís Amorim was funded by Coordenação de Aperfeiçoamento de Pessoal de Nível Superior (CAPES), which provided a master fellowship. The funders had no role in study design, data collection and analysis, decision to publish, or preparation of the manuscript.

### Grant Disclosures

The following grant information was disclosed by the authors:
Fundação de Amparo à Pesquisa e Inovação do Espírito Santo (FAPES): #036/2019.
Conselho Nacional de Desenvolvimento Científico e Tecnológico (CNPq): #80600417/17.
Renova Foundation—Brazil through its Technical-Scientific Cooperation Agreement n° 30/2018 with FEST—Brazil.
Coordenação de Aperfeiçoamento de Pessoal de Nível Superior (CAPES).

### Competing Interests

The authors declare that they have no competing interests.

## Author Contributions

- Laís Amorim conceived and designed the experiments, performed the experiments, analyzed the data, prepared figures and/or tables, authored or reviewed drafts of the article, and approved the final draft.
- Lara Chieza conceived and designed the experiments, performed the experiments, analyzed the data, prepared figures and/or tables, authored or reviewed drafts of the article, and approved the final draft.
- Jake A. Lasala analyzed the data, authored or reviewed drafts of the article, and approved the final draft.
- Sarah de Souza Alves Teodoro analyzed the data, authored or reviewed drafts of the article, and approved the final draft.
- Wesley D. Colombo analyzed the data, prepared figures and/or tables, authored or reviewed drafts of the article, and approved the final draft.
- Ana Carolina Barcelos conceived and designed the experiments, performed the experiments, authored or reviewed drafts of the article, and approved the final draft.
- Paula Rodrigues Lopes Guimarães performed the experiments, analyzed the data, authored or reviewed drafts of the article, and approved the final draft.
- João Luiz Guedes da Fonseca analyzed the data, authored or reviewed drafts of the article, and approved the final draft.
- Ana Claudia Jorge Marcondes analyzed the data, authored or reviewed drafts of the article, and approved the final draft.
- Alexsandro Santos analyzed the data, authored or reviewed drafts of the article, and approved the final draft.
- Sarah Vargas conceived and designed the experiments, authored or reviewed drafts of the article, and approved the final draft.

## Animal Ethics

The following information was supplied relating to ethical approvals (*i.e.*, approving body and any reference numbers):

The study was approved (#07/2019) by the Ethics Committee on the Use of Animals (CEUA) of the UFES. When obtaining tissue samples for genetic analyses, sampling of loggerhead turtles was performed by minimizing animal suffering. Access to Genetic Heritage was registered under SisGen (#A32C980, #A622566 and A2E6361).

## Field Study Permissions

The following information was supplied relating to field study approvals (*i.e.*, approving body and any reference numbers):

Field experiments were approved by the Ministério do Meio Ambiente—MMA.

Instituto Chico Mendes de Conservação da Biodiversidade—ICMBio Sistema de Autorização e Informação em Biodiversidade—SISBIO (License numbers: 60690-1 and 60690-2).

## Data Availability

The raw data is available in the Supplemental File.

## Supplemental Information

Supplemental information for this article can be found online at http://dx.doi.org/10.7717/peerj.18714#supplemental-information.

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
