# Peer review of "Reproductive strategies in loggerhead sea turtle Caretta caretta: polyandry and polygyny in a Southwest Atlantic rookery"

_PeerJ, doi:10.7717/peerj.18714_

## Round 0.1 · original submission · Major Revisions

Dear Authors,

Thank you for submitting your manuscript to PeerJ. After a thorough review, it has become clear that substantial revisions are necessary, particularly in the experimental design section. There are several areas where improvements are required. I look forward to receiving your revised version and encourage you to carefully consider these points to strengthen your manuscript. Should you have any questions or need further clarification, please feel free to reach out.

Sincerely,

Armando Sunny

·

Basic reporting

The work complies with the aspects considered in this section.
I can't comment on the quality of the English.

Experimental design

The work complies with the points considered in this experimental design section

Validity of the findings

All the the data is original and relevant in the field of sea turtle mating systems. There are only a few minor suggestions (see additional comments).
The conclusions of the work are well supported by the genetic information obtained

Additional comments

The manuscript presents interesting results, particularly the report on polygyny. Only a few minor changes are suggested and are mentioned below for each line number

69
Refers to within a breeding season or in consecutive years, it is suggested to clarify

165-166
Although it is understood later, it is suggested to specify that they are natural nests and not from a hatchery

177
How was the number of broods to be collected per nest decided? Use the algorithm suggested by DEWOODY, J. A., DEWOODY, Y. D., FIUMERA, A. C., & AVISE, J. C. (2000). On the number of reproductives contributing to a half-sib progeny array. Genetics Research, 75(1), 95-105.

178
How can it be ensured that the residual yolk has tissue from the brood and not just from the female?

179
It would be interesting to evaluate whether the offspring that died in the nest correspond to the genotype of a single male? That is, is there differential fitness on hatchlings associated with the paternal genotype?

217
Is it suggested to indicate the fluorophore of the marker Rox or Liz?

267
The female SMV141 is the only one that is detected in two consecutive years and it happens to have a broken carapace. It is necessary to discuss this coincidence and, if possible, have photos and more explanation about the broken carapace.

289
It is suggested to indicate, perhaps in the Excel file of the offspring genotypes,
the type of sample of the offspring: buccal sample, residual yolk or dead embryos. This information will allow us to explore whether there is any relationship between the offspring that did not hatch vs. those that did, and to detect paternal genotypes associated with low fitness from the early stages of development of the offspring.

294
Is there a relationship between sample size and multiple paternity? It is suggested to explore and mention it in the manuscript

335
Mention the number of nesting females per year at the sampling site

343
Please explore the relationship between the number of offspring analyzed per nest and the number of males identified in each nest and discuss the result

355
In order to establish this, it would be necessary for all the females compared to be in their first nest of the season. Buying a female in her second nest of the season with another in the first does not make sense if one considers that there is the capacity to sperm storage at least in one reproductive season see http://dx.doi.org/10.1016/j.jembe.2017.05.015

368
Mention the impact of having several nesting seasons analysed, as well as the number of samples and the size of the nesting female population.

381
Mention the relevance of these studies for captive breeding programs. https://doi.org/10.1016/j.gecco.2020.e01278

400
Discuss the effect, not only of the number of microsatellite loci, but also considering the number of alleles of each of them, as well as the distribution of their relative frequencies in the population.

442
This argument can be improved, if it is considered in comparison with other nesting colonies of this species with the number of nesting females per year in each rookery.

It is worth explaining then what happens with females that do not have multiple paternity, can it be related to any phenotypic attribute analyzed, or with the date of collection of samples in the field (beginning or end of the season)?

510

Including mechanisms such as cryptic female choice, see Firman, R. C., Gasparini, C., Manier, M. K., & Pizzari, T. (2017). Postmating female control: 20 years of cryptic female choice. Trends in Ecology & Evolution, 32(5), 368-382

529
Including the number of nest of the season analyzed for each female (the firs or the last)

Figure 2

It is suggested to eliminate the data of the sampled offspring (which is not relevant in the graph) and leave only the analyzed offspring, this will allow to see the information more clearly.

It is necessary to generate a different color pattern or plot between the number of males per nest, the colors used are not easy to distinguish a free tool can be found at https://colorbrewer2.org/#type=sequential&scheme=BuGn&n=3

Table 1
It is suggested to indicate the female that spawned in two consecutive years 2018 and 2020 by bold or *

Supplementary table 2

There are several females that mated with several males to describe, possibly using color code can help illustrate it on the table

eg female SMV165 is related to male 1 and 35

SMV141 in 7 and 21

SMV692 in 7, 8 and 11

SMV144 in 8 and 11

SMV154 in 21 and 22

SMV160 in 22 and 28

Reviewer 2 ·

Basic reporting

This is a solid study documenting the extent of multiple paternity in a rookery of loggerhead turtles in Brazil. It is an important contribution to the field, and I fully support it for publication. The manuscript is generally well-written, though a few sentences could be rephrased for clarity, which I have marked individually. The reference list is thorough with respect to the target species, but the discussion could benefit from additional references from studies on other species to better illustrate the broader dynamics of this reproductive strategy in marine turtles. The figures and tables are of good quality and clearly support the text.

My main critique is that while the article focuses on multiple paternity — as stated in the objectives: "the main objectives of this study were to test for multiple paternity in loggerhead nests in a rookery along the Atlantic coast of Espírito Santo, Brazil, and to estimate the breeding sex ratio for this population" — it does not explain the reason for correlating female size with the number of fathers, which appears suddenly in the methods. This aspect should be included in the research questions or objectives to provide better context.

Experimental design

The experimental design has some details that need to be clarified:
1. The sample size is confusing, the abstract says that 1109 hatchlings were analyzed but in the results section only 534 were successfully amplified which are the ones that actually contribute to the study. The methods need to be clearer as it says 20 hatchlings were selected but in Figure 2 the number of hatchlings sampled sometimes is up to 60 or 70, this is confusing. It is also important to explain if the 20 hatchlings were randomly selected or if they were the first ones to emerge? the position in the nest might be favoring the early emergence of hatchlings that might come from a particular male, for this reason randomization is important.

2. My main issue is that I roughly calculated around 17 nests that have less than 10 hatchlings genotyped which are in fact the only ones that really contribute to this study as the main objective is to calculate multiple paternity. This is problematic because with less than 10 hatchlings successfully genotyped you might be missing a larger number of fathers contributing to the nest. It doesn´t affect the overall estimate of fathers because multiple paternity is widely spread but it will definitely affect the correlation of number of fathers and female size. I suggest you limit this analysis to only nests where 15-20 hatchlings were successfully genotyped, or look for a test that could take into account the variability in the sample size of each nest, as nests with only 3 hatchlings will reduce the statistical power to detect a true correlation and might introduce variability or bias.

3. In the genetic analysis the use of the M13 tag is not specified and it adds a bit of confusion to the description of the PCR conditions.

Validity of the findings

The main findings are sound, this is an important study that shows the extent of multiple paternity and the breeding sex ratio in this loggerhead turtle rookery in Brazil. The genetic analysis was robust enough to prove this and it is an important contribution to the field. The only part of the study that lacks support is the correlation of the female size with the number of fathers in each nest due to irregular sample sizes in each nest, as this is not the main objective of the study, this is not a big issue but the analysis should be improved.

Additional comments

Individual detailed comments:

123-125 Rephrase for better understanding to: More specifically, the main objectives of this study were to test for multiple paternity in loggerhead nests in a rookery along the Atlantic coast of Espírito Santo, Brazil, and to estimate the breeding sex ratio for this population. Nothing here is mentioned about the correlation with female measurements, you need to establish all your research questions from the beginning.

173 How do you ensure that the nests were not invaded by other turtles later that night or in the following weeks?

177 The methods need to be clearer as it says 20 hatchlings were selected but in Figure 2 the number of hatchlings sampled sometimes is up to 60 or 70, this is confusing. It is also important to explain if the 20 hatchlings were randomly selected or if they were the first ones to emerge? the position in the nest might be favoring the early emergence of hatchlings that might come from a particular male, for this reason randomization is important.

194 It took me a while to understand that the "Shamblin" primers had the M13 tag, this needs to be explained in the methods because otherwise it is difficult to understand what do you mean when you say that you added "the fluorescence", this is incorrect because you add a fluorescent M13 universal tag or primer.

200 correct dNTPs, and substitute "of each primer" instead of "for each primer"

201 substitute "each fluorescent M13 tag" instead of "fluorescence for each marker"

222 Rephrase to: A subset of samples (10%) was replicated to assess the genotyping error rate. Question? Did you re-run the same PCR reaction or was the PCR reaction replicated independently for the 10% of the samples? It is important to clarify this.

224-227 the paragraph needs a bit of editing is a bit chaotic.

240 were "obtained"

267 The sentence: "Measurements were taken from 41 females, except for SMV141, for which measurements were" sounds repetitive, substitute with: "Measurements were taken from 41 females, except for SMV141, due to a carapace rupture."

270-271 check the Font of the text

288 This needs to be corrected in the abstract as it is misleading, you are saying that you analyzed 1109 hatchlings when the results come from 534, almost halfhalf of that.

294 This is my main issue, I roughly calculated around 17 nests that have less than 10 hatchlings genotyped which are in fact the only ones that really contribute to this study as the main objective is to calculate multiple paternity. This is problematic because with less than 10 hatchlings successfully genotyped you might be missing a larger number of fathers contributing to the nest. It doesn´t affect the overall estimate of fathers because multiple paternity is widely spread but it Will definitely affect the correlation of number of fathers and female size. I suggest you limit this analysis to only nests where 15-20 hatchlings were successfully genotyped, or look for a test that could take into account the variability in the sample size of each nest.

339-344 Nests with only 3 hatchlings Will reduce the statistical power to detect a true correlation and might introduce variability or bias. As I said in my previous comment I roughly counted around 17 nests with few hatchlings successfully genotyped. I suggest you limit this analysis to only nests where 15-20 hatchlings were successfully genotyped, or look for a test that could take into account the variability in the sample size of each nest.

366 Nothing is said about gene Flow, I would eliminate it

390 Rephrase to: "While informative, these methods are not feasible for many researchers and often rely on random chance."

442-445 To enrich the discussion I would suggest to compare with more studies of multiple paternity from each species in different locations in order to see the extent of this reproductive strategy in all marine turtles

470 Eliminate "The" and start with "Figure 5"

473-475 How does the maximum number of fathers compares with other species of sea turtles?

485-487 Rephrase sentence or review the position of the citation as it is not clear whether "The results of this study" are from this study or from Hays, Shimada & Schofield.

525-531 These analyses need to be reviewed in a way that varying sampling sizes in each nests do not affect the correlation.

---

## Round 0.2 · accepted · Accept

Dear Author's

On behalf of the editorial team at PeerJ, we are pleased to inform you that your manuscript, titled "Reproductive strategies in loggerhead sea turtle Caretta caretta: polyandry and polygyny in a Southwest Atlantic rookery," has been accepted for publication.

Congratulations again on the acceptance of your work. We look forward to seeing the impact your research will have on the field and to future submissions from you and your collaborators.

Thank you for choosing PeerJ as the journal to publish your research.

Best regards,

Armando Sunny

·

Basic reporting

no comment

Experimental design

no comment

Validity of the findings

no comment

Additional comments

After reviewing the responses to the suggestions made to the first version, as well as the changes made to the original document, I consider the article ready for publication.

Reviewer 2 ·

Basic reporting

No comment

Experimental design

No comment

Validity of the findings

All the points addressed by me previously were resolved accurately so I have no further comments regarding the experimental design or the results.

Additional comments

I am satisfied with the resolution of everyone of the points I raised, I am happy to recommend this paper for publication.